# Segmentation of HE-stained meningioma pathological images based on pseudo-labels

**Chongshu Wu**[1,4☯], **Jing Zhong**[2☯], **Lin Lin**[3], **Yanping Chen**[2], **Yunjing Xue**[3], **Peng Shi**[1,4]*

**1** College of Computer and Cyber Security, Fujian Normal University, Fuzhou, Fujian, China, **2** Radiology and Pathology Department, Fujian Provincial Cancer Hospital, Fuzhou, Fujian, China, **3** Radiology Department, Fujian Medical University Union Hospital, Fuzhou, Fujian, China, **4** Digit Fujian Internet-of-Things Laboratory of Environmental Monitoring Fuzhou, Fujian, China

☯ These authors contributed equally to this work.
* pshi@fjnu.edu.cn

**Data Availability Statement:** Data cannot be shared publicly now because of the ethics agreement, and it will be made public in the future. Data are available from the Fujian Medical University Union Hospital (contact via http://www.fjxiehe.com/) for researchers who meet the criteria

## Abstract

Biomedical research is inseparable from the analysis of various histopathological images, and hematoxylin-eosin (HE)-stained images are one of the most basic and widely used types. However, at present, machine learning based approaches of the analysis of this kind of images are highly relied on manual labeling of images for training. Fully automated processing of HE-stained images remains a challenging task due to the high degree of color intensity, size and shape uncertainty of the stained cells. For this problem, we propose a fully automatic pixel-wise semantic segmentation method based on pseudo-labels, which concerns to significantly reduce the manual cell sketching and labeling work before machine learning, and guarantees the accuracy of segmentation. First, we collect reliable training samples in a unsupervised manner based on K-means clustering results; second, we use full mixup strategy to enhance the training images and to obtain the U-Net model for the nuclei segmentation from the background. The experimental results based on the meningioma pathology image dataset show that the proposed method has good performance and the pathological features obtained statistically based on the segmentation results can be used to assist in the clinical grading of meningiomas. Compared with other machine learning strategies, it can provide a reliable reference for clinical research more effectively.

## Introduction

In recent years, tumor morbidity and mortality have increased rapidly in the global population [1]. Early detection and accurate diagnosis can make treatment more effective and thereby increase the chances of survival, and the analysis of tumor histopathological images is the gold standard for tumor diagnosis [2]. Hematoxylin-eosin (HE) staining is one of the most commonly used techniques for observing pathological paraffin sections, especially in the analysis of microscopic histopathological images of tumor tissue [3], in which the nucleus is stained with hyacinthine by alkaline hematoxylin while the cytoplasm is stained red by acidic eosin. In this way, the differences between the various cell structures in the tissue are effectively magnified. In HE-stained histopathological images, the morphology of the nucleus is the basis for

for access to confidential data. We also have uploaded the minimal underlying data set and the related specification as Supporting information files, including all original images which were processed and shown in the figures of the paper. Our research group is also building a full test dataset used in the experiments of the paper, which is under reviewing by the local ethics committee. The dataset will be uploaded online to public once the application is approved. Before putting the whole dataset online, all readers can ask for original images directly by contacting the corresponding author.

**Funding:** This work was supported by the Fujian Science and Technology Innovation Joint Fund (2018Y9112) and the Fujian Health Science Research Talent Training Project (2019-ZQN-17). The funders had no role in study design, data collection and analysis, decision to publish, or preparation of the manuscript.

**Competing interests:** The authors have declared that no competing interests exist.

judging the nature of cancer. For example, the size, shape, and density of the nucleus affect the qualitative analysis of the tumor. However, it is difficult for pathologists to analyze massive amounts of HE-stained data, and the results of the analysis are susceptible to subjective factors [4].

Researchers are seeking breakthroughs in the segmentation of nuclei in pathological images because nuclear detection and segmentation are the most basic and critical steps in the process of pathological image analysis. Most of morphological methods are based on the threshold, watershed algorithm, statistical level set, active contour model, or a combination of these approaches [5–9]. These methods provide excellent segmentation results under certain conditions. However, in the actual pathological image analysis of HE-stained tissues, the following challenges may be encountered when using the above methods. First, because HE staining is greatly affected by external factors, there is extensive variability in staining nuclei. Second, the lack of a clear nuclear boundary makes these methods extremely prone to over-segmentation. Third, the diversity in nuclear shapes makes it difficult to establish a stable shape model.

With the rapid development of machine learning, image segmentation is simplified as labeling all image pixels through learning, which consist of two major categories: 1) Classic machine learning segmentation methods: The segmentation of pixels is based on feature sets extracted based on well-designed models, and inputted into classic machine learning methods to assign a certain category to each pixel in the whole image. 2) Deep learning methods for image segmentation: Various typical image patches are manually selected as the training dataset, and inputted into a neural network model to segment different cell parts such as nucleus, cytoplasm and extracellular space (ECS), then the trained network is used to segment all other unmarked images.

As classic machine learning approaches, Mittal et al. [10] used super-pixel clustering method to segment cell nuclei, and used gravity search algorithm to optimize cluster centers. Qu et al. [11] proposed a method based on the pixel-wise support vector machine (SVM) classifier for segmenting tumor nests and the stroma. Meanwhile, recent breakthroughs have been made in the analysis of natural and pathological images with deep learning, various methods of nuclear segmentation based on deep learning have emerged. In segmentation tasks, initially, convolutional neural networks (CNNs) were used only for feature extraction, and with the emergence of fully convolutional networks (FCNs) [12], semantic segmentation has gradually become a mainstream segmentation method, and various improved models have emerged. Chen et al. [13] presented a novel deep contour-aware network (DCAN) and used a multilevel contextual FCN to generate multiscale feature representations in an end-to-end manner. Through the modification of the FCN model, Qu et al. [14] obtained a full-resolution convolutional neural network (FullNet) to retain more detailed information. The U-Net Ronneberger et al. [15] proposed brought new vitality to the segmentation model of pathological images, and many optimized versions were produced. Pan et al. [16] extended U-Net with atrous depthwise separable convolution (AS-UNet) for nuclear segmentation. Study [17] proposed a U-Net-based neural network residual channel attention U-Net (RIC-UNet), which applied residual blocks as well as multiscale and channel attention mechanisms to RIC-UNet to enhance the segmentation accuracy.

The above methods are all improvements in terms of the model aspects, but one of the essential problems of deep neural networks is still unsolvable: the necessity of a large number of manually labeled data points, which is especially important in medical image analysis. Since there is high requirement for professional in medical image annotation, to further improve the efficiency of machine learning in the training dataset preparation. In this regard, we propose a fully automatic pathological image nuclear segmentation method based on pseudo-label as shown in Fig 1. First, K-means clustering based on pixel features is performed on the images,

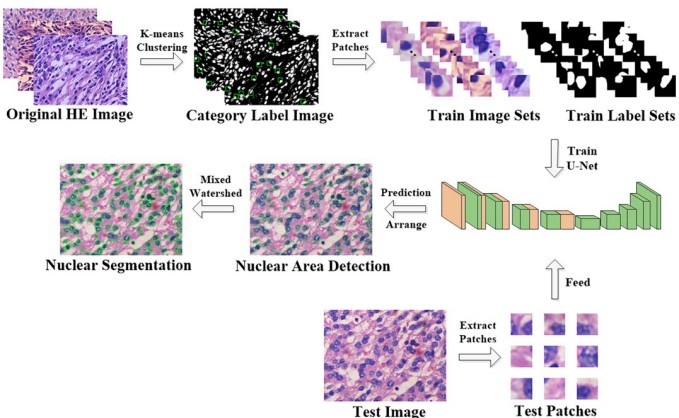

**Fig 1. Workflow of the proposed HE image processing and analysis pipeline.**

and then the training sample patches are grabbed according to both the established rules and the clustering results. Second, a novel training algorithm that is not fixed epoch is designed, and the obtained training sample patches are put into the model for training with the full mixup operation according to the probability. Then, overlapping patches of the test images are cut into the same size and delivered to the model to obtain the prediction results, and the foreground segmentation results of the nuclei are obtained by stitching. Finally, the hybrid watershed algorithm is used to segment the boundaries between nuclei, the final segmentation result is obtained and the relevant pathological features are counted. Comparing to both traditional machine learning and deep learning methods, the use of unsupervised clustering replaces the tedious process of manually annotating the training sets, and the full mixup strategy is used to solve the problem of the poor generalization ability of the automatically acquired training patch sets.

## Materials and methods

In this section, we mainly focus on the detailed process of training samples and their pseudo-labels, which can automatically grab reliable patches from complete HE-stained images. To solve the generalization problem, a series of training and testing protocols are proposed.

### Acquisition of HE-Stained images and nuclear annotation

In this work, we used meningioma pathological images as the experimental subjects. The World Health Organization classifies meningiomas into three grades: benign meningiomas (WHO I), atypical meningiomas (WHO II), and interstitial or malignant meningiomas (WHO III). The first grade is known as the low-grade, and the second and third grades are categorized as high-grade meningiomas [18]. Therefore, there are certain differences between high- and low-grade meningiomas, which are highly suitable for testing the generalization ability of the proposed method.

Dataset preparation was performed on tissue samples of two groups including high- and low-grade meningiomas from different clinical patients in Fujian Medical University Union Hospital. The local ethics committee granted ethical approval for the study (Certificate No. 2019KJTYL024), and informed consent was obtained. Microscopic slices were stained by HE with the standard histological procedure described in [19], and then RGB images were captured and stored as $1536 \times 2048 \times 24$-bit tiff files. Eventually, we obtained a total of 60 HE-

stained tissue sections of high- and low-grade meningiomas, including 30 high-grade and 30 low-grade meningiomas. Images of five most representative areas determined by professionals in each slice were collected to form HE image sets. Therefore, a total of 300 (60*groups* × 5) HE-stained images were used in this work.

To include as diverse of nuclear appearances as possible, we randomly selected 20 images of different groups from the collected 60 groups including 300 HE-stained pathological images captured from both high- and low-grade meningioma samples. The selected dataset included 10 high-grade and 10 low-grade meningioma images. A 1536 × 2048 HE-stained pathological image has approximately 800 nuclei to be annotated, so there are about 16,000 nuclei in 20 images. We used the software MaZda4.6 [20] to annotate these nuclei, and the annotators were two doctors from Fujian Medical University's Union Hospital.

## Training sample selection process based on unsupervised learning

This section introduces a method of unsupervised selection of training samples, which automatically select most reliable patches including nuclei and their labels based on dyeing characteristics of each slice. These samples are then used to train the deep learning model for nuclei segmentation of other HE-stained images.

**Feature selection.** There are two kinds of quality problems in HE-stained images: uneven distribution of dyes in tissues, and salt and pepper noise caused by small dye particles. According to these problems, the image is filtered by a combination of a 5 × 5 median filter [21] and a 5 × 5 Gaussian filter [22] to eliminate noises. Earlier studies of our group have proven that the selected feature sets are effective for segmentation of the nuclei [23]. The two-color channels R and G provide much more information for the classification result than the B color channel. Therefore, to reduce computational cost and improve the effectiveness of the proposed pipeline, intensities of color channels R and G are selected and mapped into the two-dimensional feature space to form a two-dimensional feature set.

**Acquisition of stable color areas of the nuclei.** Manhattan distance-based K-means clustering is applied to automatically obtain the nuclear stable color areas. There are three different cell structures in the HE-stained images, namely, the nuclear area, the cytoplasmic area and the ECS. There is no clear boundary between the above structures in the HE-stained images. To obtain more reliable nuclei areas for building the training set, we firstly set the clustering classes as five and randomly initialize clustering centers. Since the eigenvectors modules of nuclei, cytoplasma and ECS feature sets are ascending in practice, the mean values of all clustering centers are also following this order. Then the following five pixel-level categories are obtained from clustering: the nuclear stable color area, the nuclear-cytoplasmic fuzzy area, the cytoplasmic stable color area, the cytoplasmic-ECS fuzzy area, and the ECS stable color area.

Only Precision needs to be considered in clustering nuclear areas for the automatic nuclear patches selection, and the increasing of Recall might give more options in selecting proper training samples. We extract several representative HE-stained images and compare the nuclear areas obtained by clustering with the artificially sketched nuclear areas. Fig 2 shows the distribution of the stable color pixels of nuclei in different images. The olive areas are the deeply stained parts of stable color areas of the nuclei obtained by clustering can be used as the candidate areas for training samples.

**Collection of training image sets.** Since our task is to segment the whole nuclei areas, we further merge the above five categories obtained by clustering. First, pixels belonging to the stable color areas of the nuclei can be used as the foreground of the training samples, so no changes are made. Second, pixels belonging to the nuclear-cytoplasmic fuzzy areas include both the nuclear and cytoplasmic pixels, so they are defined as fuzzy areas. Finally, pixels

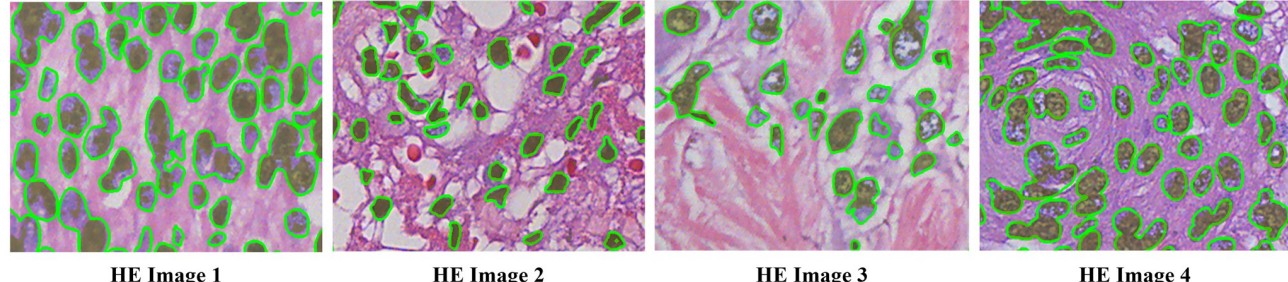

**HE Image 1**   **HE Image 2**   **HE Image 3**   **HE Image 4**

**Fig 2. Comparison of the ground truth and clustering results of nuclei.** Where the olive areas indicate the stable color areas of the nuclei in the clustering result, and the green outlines indicate ground truths of the nuclear stable color area.

**Table 1. Comparison results before and after integration of each category.**

| Clustering results | Integration results |
| --- | --- |
| the nuclear stable color area | the nuclear stable color area |
| the nuclear-cytoplasmic fuzzy area | fuzzy areas |
| the cytoplasmic stable color area | the nonnuclear area |
| the cytoplasmic-ECS fuzzy area | |
| the ECS stable color area | |

belonging to the cytoplasmic stable color areas, the cytoplasm-ECS fuzzy areas, and the ECS stable color areas are combined into one category, which is defined as the nonnuclear area. Detailed integration results are demonstrated in Table 1.

The automatic patch collection is shown as pseudocolor images of the different categories in Fig 3. The training image sets are built based on the following screening rules. We design a window with a size of 48 × 48 pixels and set the stride size to 8 pixels to slide on the entire category label pseudocolor image. When more than 100 nuclear stable color area pixels are in the window and the proportion of pixels in the fuzzy area is less than $\alpha$, the windowed original HE image is captured as the training image patch, and the corresponding position in the pseudocolor image of the category label is captured as the label of the training image patch. Since the fuzzy areas are basically around the nuclear stable color area, it is difficult to completely avoid the fuzzy areas. To minimize the impact of the fuzzy areas on the training effect, we set $\alpha$ to 0.05 here. The result is shown in the training image patch and training label patch in Fig 3.

## Training and testing protocols

As shown in Fig 2, since the nuclei that can be captured by unsupervised are the nuclei in each image and basically do not exceed the manually annotated contour position of green lines,

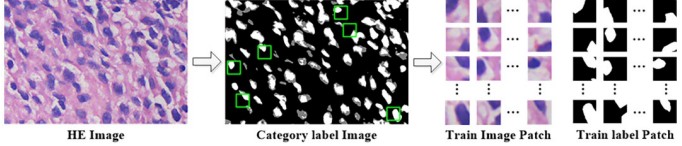

**HE Image**   **Category label Image**   **Train Image Patch**   **Train label Patch**

**Fig 3. Training image patch collection process.** Where the white areas in the category label image are the nuclear stable color areas, the gray areas are the fuzzy areas, the black areas are the nonnuclear areas, and the green boxes indicate the areas selected as the training.

which shows that the most deeply stained parts, the training sets collected by automatic collection are directly used for the model training, which may cause under-segmentation of the nuclei with light staining. Usally it is difficult to distinguish some light blue nuclei with the surrounding cytoplasma. Therefore, accurate segmentation of the lightly stained nucleus is the key challenge of nuclear segmentation. We design a series of training and testing protocols for the training sets to improve the generalization ability of the proposed model to maximize the resolution of the lightly stained nuclei.

**Convolutional neural network model.** We use the U-Net [15] network model, which is one of the most widely used networks in image segmentation, to verify the proposed pseudo-label construction method. The U-Net model is an improvement and extension of the FCN, which use convolution layers and pooling layers for feature extraction and then use deconvolution layers to restore the image size. The model is a mature baseline semantic segmentation model, and its effectiveness has been fully verified. The simple structure of U-net can minimize the impact of the segmentation model itself, and verify the proposed pseudo-label method more precisely.

**Training and testing sets.** We use 20 manually annotated images as the testing sets of the model. Considering that the five images of the same group acquired from one patient at the same time, the nuclear morphology and staining were highly similar. Therefore, to ensure the reasonableness and reliability of the test accuracy, the images used as the test sets and other images in the same group are not involved in the production of the training sets. Using the designed unsupervised method to find a reliable training set, more than 200,000 patches with a size of $48 \times 48$ that can be used for training are obtained from the remaining 40 groups of 200 images. We allocate the training sets and the validation sets according to a ratio of 4 to 1, so the training sets generally contain more than 160,000 patches and the validation sets have more than 40,000 patches.

**Training patch full mixup.** Because the training sets we constructed are composed of the most well-stained nuclei in different HE-stained images, the network trained with the original training sets cannot segment the lightly stained nuclei, resulting in a lack of generalization ability. A full mixup strategy of training samples is applied to compensate this problem. We use the following methods to perform the mixup operation:

$$I_{mix} = \lambda I_x + (1 - \lambda)I_y \tag{1}$$

$$L_{mix} = L_x | L_y \tag{2}$$

where $I_x$ and $I_y$ are different input training image patches and $L_x$ and $L_y$ are different input training label patches. As the Eqs (1) and (2) shows, the full mixup operation merges different images with λ as the weight and directly superimposes the labels of the images. Because each HE-stained image is real and effective, they have the same importance, so we set the mixing weight λ to 0.5. 'Or' operation is performed between binary labels of $L_x$ and $L_y$ in Eq (2).

As shown in Fig 4, comparing with the original image, there are many more lightly stained nuclei in the mixed image, which is caused by the mixing of the nuclear areas of one image with the cytoplasmic areas or ECS areas of another image. The nuclei generated by the image mixing are highly similar to the lightly stained nuclei in the original image, so this operation fixes the problem that only the mostly stained nuclear areas could be selected in self-supervised learning. In practice, we determine whether a group of batch-size training sets in the input network needs to be fully mixed according to the probability $P$. If necessary, this set of training sets is randomly interleaved and fully mixed within the group, and $P$ is variable while training is in progress.

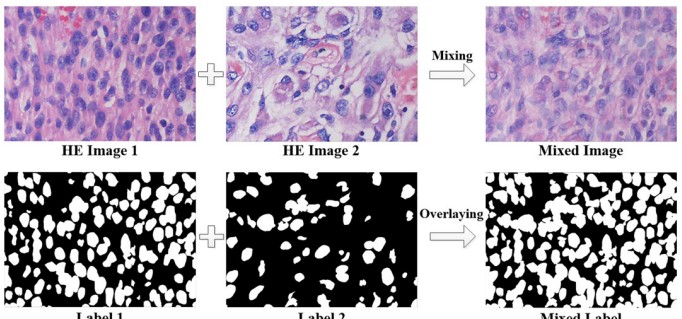

**Fig 4. Full mixup operation diagram.** The first line is the mixing process of HE images, and the second line is the mixing process of labels.

**Loss function.** The training effect of the training set image after full mixup is the key to whether the model can distinguish lightly stained nuclear pixels. To increase the penalty intensity for the nuclear pixels in the mixed image, we use the weighted binary cross entropy loss function, and the equation is as follows:

$$Loss = mean(l_1, l_2, \ldots, l_N) \tag{3}$$

among them,

$$l_n = -w_n[y_n \cdot logx_n + (1 - y_n) \cdot log(1 - x_n)] \tag{4}$$

where $N$ is the batch size, $y_n$ is the ground truth value of the nuclear image, and $x_n$ is the predicted value (between 0 to 1). For the weight $w_n$, which is initialized as a matrix with all elements of 1, we impose twice the weight on the nuclear pixels in the batch if the full mixup is performed, but for the background and the batch without the full mixup operation, we maintain the original weight.

**Evaluation criterion.** Criterions including accuracy (ACC), precision (PC), recall (RC), specificity (SP) and F1-score (F1) are widely used to evaluate the performance of nuclear segmentation in pathological images, and the calculation equations for these parameters are shown in Eqs (5)–(9).

$$ACC = \frac{TP + TN}{TP + FP + FN + TN} \tag{5}$$

$$PC = \frac{TP}{TP + FP} \tag{6}$$

$$RC = \frac{TP}{TP + FN} \tag{7}$$

$$SP = \frac{TN}{TN + FP} \tag{8}$$

$$F1 = \frac{2 \times PC \times RC}{PC + RC} \tag{9}$$

The accuracy (ACC) represents the ratio of correctly segmented nuclear and background pixels to the total number of pixels in the image. In the comparison of the segmentation results

(SR) with ground truth (GT), the precision (PC) represents the proportion of correctly segmented nuclear pixels in SR to the total nuclear pixels in SR. The recall (RC) can also be called the sensitivity, which is relative to the specificity (SP). The former indicates the ratio of correct nuclear pixels in SR to the nuclear pixels in GT, and the latter indicates the ratio of correct nonnuclear pixels in SR to nonnuclear pixels in GT. The F1-score(F1) value is the harmonic mean of the precision and recall, and TP, FP, FN and TN represent the true positive, false positive, false negative and true negative, respectively. In addition, we introduce two other indicators from [24], the Jaccard similarity (JS) and the Dice coefficient (DC), and their calculation equations are shown in Eqs (10) and (11).

$$JS = \frac{|GT \cap SR|}{|GT \cup SR|} \tag{10}$$

$$DC = 2\frac{|GT \cap SR|}{|GT| + |SR|} \tag{11}$$

From the Eqs (10) and (11), we can see that the JS and DC both describe the coincidence of the SR and GT; these parameters are often used to calculate the similarity or overlap of the SR and GT. When SR and GT completely coincide, the values of JS and DC are equal to 1.

**Dynamic training strategy.** During model training, unlike the commonly used fixed values of epochs, we use the probability value $P$ in the full mixup operation to determine the training epoch. The pseudocode of the dynamic process is described as Algorithm 1:

**Algorithm 1** Training strategy

```
Input: JS and DC indicators in the training accuracy and validation
       accuracy of each epoch;
Output: The best U-Net model weights;
1: Set and initialize the parameters: Best_score ← 0; P ← 0.1;
2: for each epoch JS_{t_i}, DC_{t_i}, JS_{v_i}, DC_{v_i} do
3:    ScoreT ← JS_{t_i} + DC_{t_i};  ScoreV ← JS_{v_i} + DC_{v_i};
4:    if ScoreV > Best_score - 0.005 and |ScoreT - ScoreV| < 0.06 then
5:      Save the current model weights;
6:      Update the parameters:Best_score ← ScoreV; P ← P + 0.1;
7:    end if
8:    if P < 1 then
9:      Continue training;
10:   else
11:     End training;
12:   end if
13: end for
```

In Algorithm 1, $P$ is the initial probability of performing full mixup; $JS_{t_i}$ and $DC_{t_i}$ are the JS and DC in the training accuracy index obtained from each epoch of training; $JS_{v_i}$ and $DC_{v_i}$ are the JS and DC in the validation accuracy index obtained from each round of training. In the 4th step of the algorithm, as the training progresses, the $P$ value increases step-by-step; that is, an increasing number of images are fully mixed, resulting in a slight fluctuation in the training accuracy. Thus, ScoreV is slightly lower than the Best_score obtained from the $P$ value of the previous stage, and the decrease range is set to 0.005. Ensuring that the distance between ScoreT and ScoreV is within a controllable range prevents the model from overfitting.

The final $P$ value ranges from 0.1 to 0.9. Here the upper limit is set as 0.9 because completely mixing the training sets results in insufficient training of the original image by the model. This step-by-step training method can allow the model to abundantly train on the original images and then train on the full mixup images, which is equivalent to a process of gradually generalizing the model.

**Prediction result stitching.** In the testing, to keep the input size of the model the same, we crop the test image into patches sized 48 × 48 and set the cropping step to 24, which allows the boundary part of each patch to be predicted multiple times and improves the stability of the model at the patch boundary. When performing segmentation result stitching, since the cropping step is 24, overlapping parts are generated multiple times during stitching. Then, we average the probability values of the corresponding overlapping parts as the final result. Finally, the pixels with a predicted probability value greater than or equal to 0.5 are used as the nuclear pixels.

## Overlapping nuclei segmentation

In clinical research, quantitative analysis of different cell morphologies in histopathological images is often required, so the simple division of nuclear areas is not adequate. Therefore, the nuclear areas need to be precisely segmented to determine the boundaries of each nucleus. However, due to the lack of delineated boundaries between the overlapping nuclei, morphological method is needed to further segment the boundaries of the nuclei in the fully automatic pipeline.

The common operation steps of the watershed algorithm are color image graying, gradient map construction, and watershed segmentation based on the gradient map to obtain the edge line of the segmented image. When the initial catchment area (i.e., the minimum value of the area) is designed, the nuclear area obtained by the foreground segmentation is subjected to distance transformation to extract the morphological center point, and the stable color areas of the nuclei obtained by the clustering together constitute the initial catchment areas. Then, based on these areas, the watershed algorithm is used to obtain the segmentation results of the adherent nuclei.

## Results and discussions

We first conducted ablation experiments on the main training strategies designed to verify the effectiveness of each strategy. Secondly, the proposed method is compared with the traditional pixel-based machine learning segmentation method to prove the advantages of the image block training set constructed by unsupervised learning. Then, in order to reflect the advantages of unsupervised construction of training samples, it is compared with the fully-supervised semantic segmentation method. Finally, the effect of the hybrid watershed method on the segmentation between cores is verified.

The study is implemented with Python 3.7, and the algorithms are developed based on the deep learning framework of Pytorch, which is a now commonly used machine learning library. The core components of the hardware environment are an Intel (R) Core (TM) i7–8700K CPU, 16 GB RAM and a Nvidia Titan RTX GPU.

### Ablation experiment performance

To verify the necessity of the series of training protocols we proposed, we use 20 original HE-stained images with artificially annotated nuclei for testing, while the training sets and validation sets are automatically captured by unsupervised learning. The designed ablation experiment and the experimental results of segmentation criterions are shown in Table 2.

The operation items in Table 2 are the training protocols in the previous section. In addition, the general mixup operation, in which both the training images and labels are mixed with random weights that conform to the beta distribution, is added for comparison with full mixup. The tick in the table indicates corresponding operations are performed in each row.

**Table 2. Ablation results of segmentation criterions.**

| Operation | | | | Segmentation criterions | | | | | | |
|---|---|---|---|---|---|---|---|---|---|---|
| Mixup | Full mixup | Weighted | Flexible epoch | ACC | PC | RC | SP | F1 | JS | DC |
| — | — | — | — | 0.9221 | 0.8850 | 0.7213 | 0.9760 | 0.7851 | 0.6504 | 0.7851 |
| ✓ | — | — | — | 0.9281 | **0.8956** | 0.7386 | **0.9785** | 0.8010 | 0.6713 | 0.8010 |
| — | ✓ | — | — | 0.9371 | 0.8591 | 0.8225 | 0.9666 | 0.8366 | 0.7201 | 0.8366 |
| — | ✓ | ✓ | — | 0.9337 | 0.8178 | 0.8616 | 0.9511 | 0.8350 | 0.7191 | 0.8350 |
| — | ✓ | ✓ | ✓ | **0.9385** | 0.8337 | **0.8800** | 0.9536 | **0.8522** | **0.7441** | **0.8522** |

From the results we can see that the ACC values of different operations are very close. This similarity occurs because in the meningioma HE-stained images, nonnuclear areas usually have a large proportion in the whole image, and it is not difficult for the general model to correctly segment most of the nonnuclear areas, which leads to the dilution of the incorrect segmentation of the nuclear areas. The PC values and SP values show downward trends because the insufficient generalization ability of the original model, which can only identify darker stained nuclei based on the training samples, and these areas are basically the correct nuclear areas, leading to a lower FP. With additional operations, the generalization ability of the model is enhanced, and can distinguish the lightly stained nuclei, causing the RC value and F1 value to increase. That indicates the generalization ability of the model is gradually increasing with multiple operations shown in the last row of Table 2, and the nuclei foreground segmentation results based on the final combined operations are shown in Fig 5.

Note that the proposed self-supervised learning method does not require the color normalization of the image before or after the training patch is collected, which allows the image to maintain the original color features and further saves the time in training set preparation.

## Comparison of the segmentation performance with traditional machine learning methods

Usually there are two mainstream traditional machine learning methods used for segmentation pathological images: pixel-level classification based on supervised SVM [6] and unsupervised hierarchical K-means clustering [25].

We designed the following experiment to compare the segmentation effect of the proposed pseudo-label method with those of traditional supervised and unsupervised methods: 1) SVM method: Since the method described in [11] needs to be based on the same group of HE-stained images, we randomly take out one percent of the total number of pixels in the image and its pixel labels for training the SVM and then use the trained model to predict each pixel of the entire image until all 20 test images are predicted and the average accuracy is calculated.

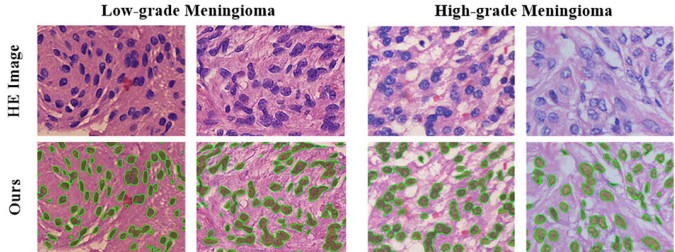

**Fig 5. Nuclear area segmentation results of low- and high-grade meningioma images based on both the manual and proposed methods.** where the olive areas are the ground truth, and the areas surrounded by thin green lines indicate our nuclear segmentation results.

**Table 3. Comparison with traditional machine learning methods.**

| Method | ACC | PC | RC | SP | F1 | JS | DC |
|---|---|---|---|---|---|---|---|
| K-Means | 0.8920 | 0.6774 | **0.9062** | 0.8928 | 0.7592 | 0.6243 | 0.7592 |
| SVM | 0.8924 | 0.7703 | 0.6233 | **0.9559** | 0.6840 | 0.5387 | 0.6840 |
| Ours | **0.9385** | **0.8337** | 0.8800 | 0.9536 | **0.8522** | **0.7441** | **0.8522** |

2) The hierarchical K-means clustering method is the same as the method described in study [25]. The index results obtained by different methods of segmentation are listed in Table 3.

For traditional machine learning segmentation method based on pixel classification, even if the local area feature of the pixel is added, there is still lack of correlations between the pixels, so this kind of method needs some morphological postprocessing to produce relatively complete nuclear areas. Meanwhile, the proposed method needs only to splice each patch, with no excessive morphological postprocessing required. The comparison results are shown in Fig 6.

Fig 6 shows the limitations of the traditional pixel-based machine learning segmentation method. The lack of local perception results in many incorrectly segmented pixels, so morphological postprocessing is required to improve the segmentation accuracy. Besides, postprocessing of the morphology can easily lead to the situation shown in Fig 6 Sample 2. The cytoplasmic area surrounded by multiple nuclear areas is mistakenly classified as nuclear areas, resulting in a decrease of nuclear segmentation accuracies.

## Comparison of the segmentation performance with supervised semantic segmentation

To construct training datasets for supervised learning, we split the 20 artificially annotated HE-stained images into two parts by 3:1, 15 for training, and the remaining 5 for testing. We

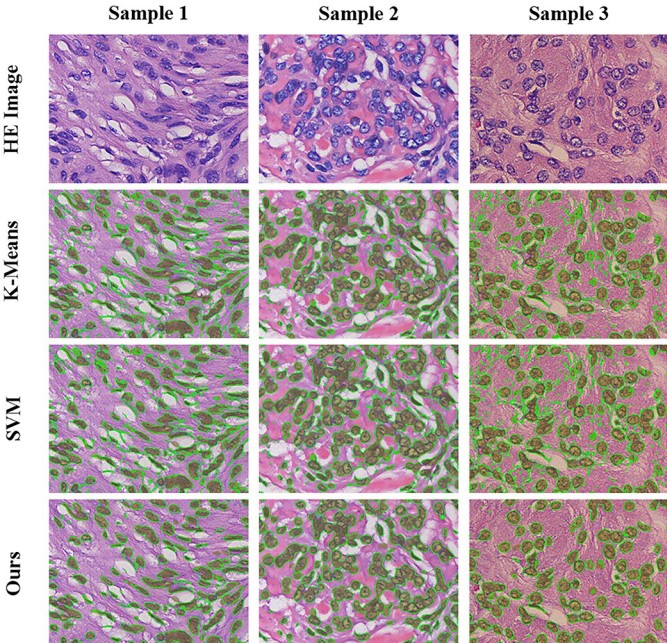

**Fig 6. Comparison of the segmentation results with those of traditional machine learning methods.** Each row represents the segmentation results of different methods, and each column represents different samples. The olive areas in the image represent ground truth, and the green line represents the result of nuclear segmentation using the corresponding method.

**Table 4. Comparison with supervised semantic segmentation methods.**

| Method | ACC | PC | RC | SP | F1 | JS | DC |
|---|---|---|---|---|---|---|---|
| U-Net | 0.8189 | 0.6950 | 0.9046 | 0.7982 | 0.7459 | 0.6195 | 0.7459 |
| AttU-Net | 0.8117 | 0.6913 | **0.9113** | 0.7867 | 0.7452 | 0.6210 | 0.7452 |
| R2U-Net | 0.8780 | 0.7592 | 0.7720 | 0.9208 | 0.7209 | 0.5826 | 0.7209 |
| Ours | **0.9250** | **0.7970** | 0.8846 | **0.9366** | **0.8356** | **0.7202** | **0.8356** |

randomly grab more than 200,000 training patches from 15 labeled images to ensure that the number of training samples is the same as that in the pseudo-label method. The training data-sets are augmented by means of left-right and up-down shifting, rotating, flipping and rescaling operations, and then the U-Net model and its two improved models are used for comparison. After sufficient training, the test images are input into all the models for prediction.

The results are shown in the Table 4, where AttU-Net means Attention U-Net [26], and R2U-Net presents the Recurrent Residual CNN-based U-Net [24]. The experimental results expose the limitations of supervised learning approached based on manual labeling, which has limited generalization ability based on the diversity of manual labeling. Since we randomly selected 20 labeled images from different groups, there are obvious differences in the color and nuclear morphology of the images. In addition, the training images are not color-normalized during training, so it is difficult for the supervised learning model to directly learn the features of the other 5 test images from the 15 training images, and that restricts the accuracies shown in Table 4. The comparison images of the segmentation results of each method are shown in Fig 7.

From sample 2 segmented by AttU-Net and sample 1 segmented by R2U-Net in Fig 6, we see that whether the introduction of an attention mechanism to strengthen AttU-Net learns the effective features or the recurrent neural network (RNN) and ResNet Structure integration of R2U-Net is used, there are still several images that cannot be effectively segmented. This failure may be due to no similar images being present in the training set, meaning the model is unable to effectively learn the features of such images. In the real HE-stained pathological image segmentation task, the quality of the images varies a lot, which makes comprehensive manual annotation data difficult to obtain. One solution is to normalize the color of the HE-stained images so that the color of the images tends to be the same. This can reduce the efficiency of batch image processing, but the selection of the reference image will directly affect the quality of the overall normalized result.

## Comparison of the number of nuclei

To precisely quantify the distributions and morphologies of single cells in the slice, we use the nuclear count method to test the segmentation of the adherent cell nuclei, which also provide statistical results for researches on clinical diagnosis or classification. We randomly select three images from the high- and low-grade meningioma test images respectively as the samples to be counted, and then send them to the pathologist for estimation of nuclear number statistics. Due to the high density of cells in the complete image, the pathologist divides each image into 16 ($4 \times 4$) areas, each with a size of $384 \times 512$ pixels, and then the expert selects four ROIs for statistics based on the cell distribution. The number of nuclei in the entire image is estimated based on the principle of uniform cell distribution. The results of the comparison between the proposed hybrid watershed segmentation method and the manual statistics are shown in Table 5, in which Manual Statistics represents the result of manually counting the

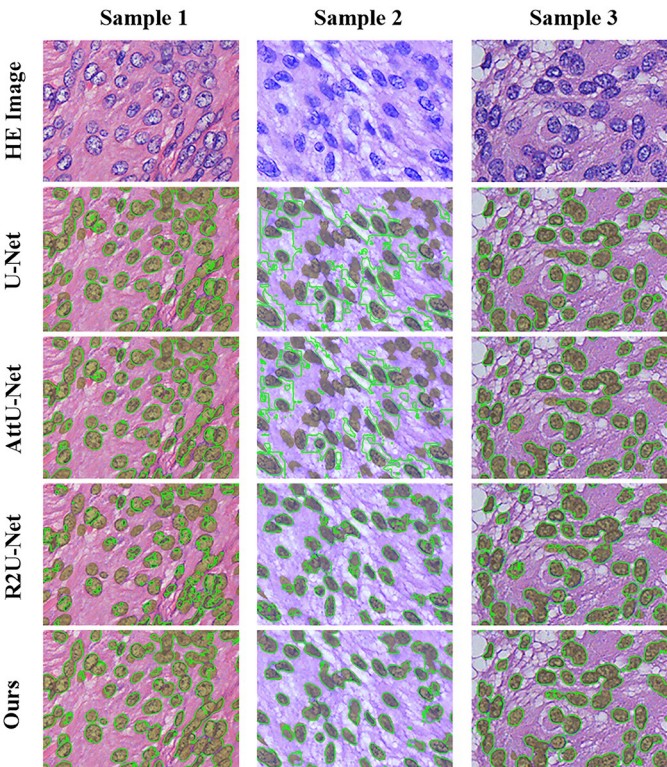

**Fig 7. Comparison of the segmentation results with supervised semantic segmentation methods.** Each row represents the segmentation results of different methods, and each column represents different samples. The olive areas in the image represent ground truth, and the green line represents the result of nuclear segmentation using the corresponding method.

number of nuclei, and Before represents the statistical results before the adhesion nuclear segmentation, that is, the result of directly counting the number of nuclear foreground areas obtained by proposed method. After represents the statistical results after the Mixed Watershed segmentation, Rate represents the rate of increase in the number of nuclei after adding the adhesion nuclear segmentation. Error gives an error range of the final statistics after watershed segmentation compared with the manual statistics.

The application of mixed watershed segmentation has a relatively strong impact on the number of nuclei, the lowest increase is greater than 25%, and most of the errors compared with the manual statistics are lower than 10%, which shows that the mixed watershed method

**Table 5. Nuclear count statistics.**

| Samples | | Manual Statistics | Mixed Watershed | | | Error |
|---|---|---|---|---|---|---|
| | | | Before | After | Rate | |
| Low-grade | 1 | 959±21 | 657 | 1010 | 53.73% | 3.06%-7.68% |
| | 2 | 904±13 | 590 | 919 | 55.76% | 0.22%-3.14% |
| | 3 | 726±15 | 480 | 774 | 61.25% | 4.45%-8.86% |
| High-grade | 1 | 897±28 | 554 | 821 | 48.19% | 5.52%-11.24% |
| | 2 | 574±20 | 445 | 557 | 25.17% | 0.54%-6.23% |
| | 3 | 904±19 | 592 | 911 | 53.89% | 1.30%-2.94% |

has the ability to segment the adherent nuclei. The detailed segmentation performances are shown in Fig 8, which shows the segmentation results of three original (1536 × 2048) HE-stained sample images. From the 2-level zoom-in views, we can see that most of the borders of the adherent nuclei are accurately found and segmented. The hybrid watershed segmentation method achieves good performance because it can select the appropriate local minimum points according to the color of the nucleus and the reliable morphology of the adhered

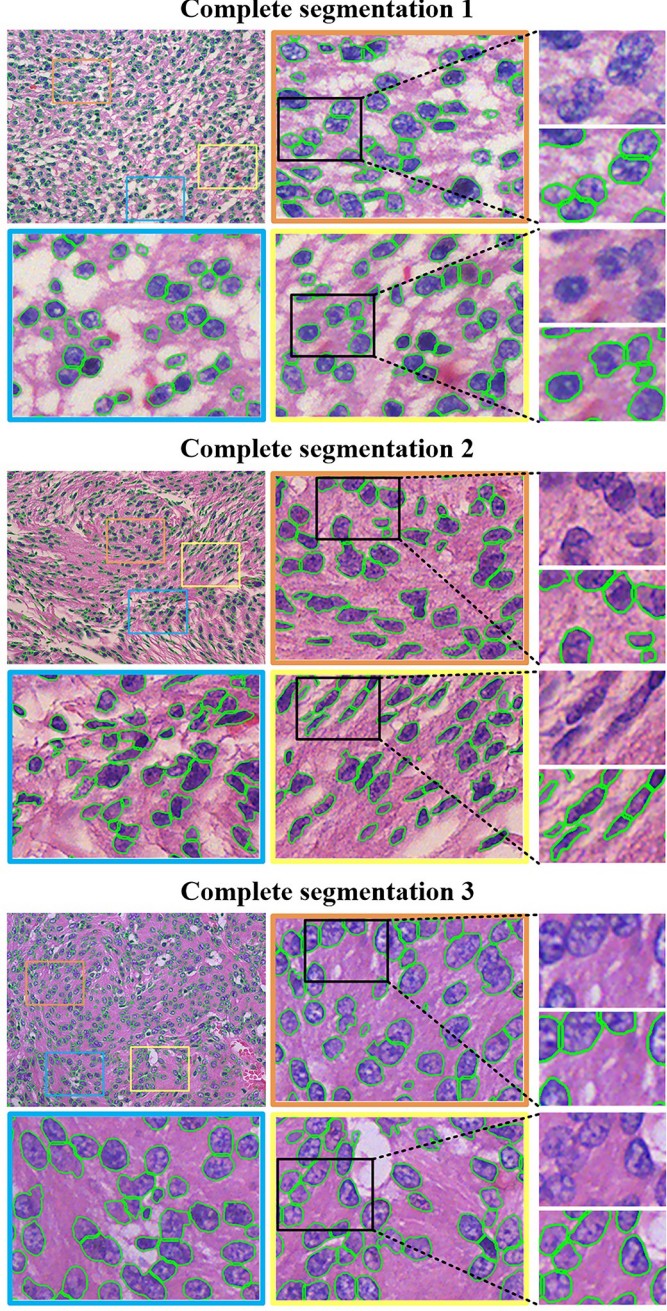

**Fig 8. Complete segmentation performance.** The images with orange, blue, or yellow borders are the results of partial enlargement; the third column shows even greater segmentation details.

**Table 6. Feature set of HE-stained meningioma pathology images.**

| Feature | Definition |
|---|---|
| Nuclei number | Number of nuclei |
| Nuclei area ratio | Nuclei area / Total image area |
| Cytoplasm area ratio | Cytoplasm area / Total image area |
| ECS area ratio | ECS area ratio / Total image area |
| Nuclei to cytoplasm area ratio | Nuclei area / Cytoplasm area |

nucleus, and then obtain the accurate boundary of the adhered nucleus. The prerequisite for this method is that the segmentation method based on pseudo-label can segment a more accurate foreground nuclear area from the HE-stained image, which provides favorable conditions for the hybrid watershed method.

## Statistical analysis of pathological features of meningioma

The quantification of the segmentation results of HE-stained images to output relevant pathological features is one of the important steps in pathological image analysis. In order to verify that the pathological features based on the segmentation results have an auxiliary effect on meningioma grading, we quantified the segmented images using a series of pathological features defined in the previous study [23], as shown in Table 6, which characterise the state of the tumour tissue and can reflect the progression of the tumour [23].

Since no manual annotations were involved in all training processes, there is no need to consider the data leakage problem in deep learning here. The proposed method for segmenting the nuclei of meningioma pathology images was applied to a total of 60 groups of 300 images (30 groups of 150 for high-grade and 30 groups of 150 for low-grade), and the cytoplasmic regions and extracellular interstitial regions were temporarily segmented using K-means clustering. Based on the segmentation results, the feature sets in Table 6 were counted and some of the results are shown in Tables 7 and 8.

The Wilcox rank sum test was used to test for differences in features between high and low grade meningiomas. The null hypothesis: there is no significant difference between the statistical characteristics of high and low grade meningiomas. The test results in the case of a confidence level of 0.95 are shown in Table 9.

**Table 7. Low grade meningioma features.**

| Sample | Nuclei number | Nuclei area ratio | Cytoplasm area ratio | ECS area ratio | Nuclei to cytoplasm area ratio |
|---|---|---|---|---|---|
| 1 | 958 | 22.49% | 31.68% | 45.83% | 71.00% |
| 2 | 987 | 20.01% | 33.30% | 46.69% | 60.10% |
| 3 | 1141 | 22.71% | 29.86% | 47.43% | 76.03% |
| 4 | 986 | 20.58% | 34.34% | 45.08% | 59.93% |
| 5 | 833 | 27.21% | 33.66% | 39.13% | 80.85% |
| ⋮ | ⋮ | ⋮ | ⋮ | ⋮ | ⋮ |
| 146 | 1122 | 16.86% | 42.96% | 40.18% | 39.24% |
| 147 | 776 | 17.47% | 47.73% | 34.80% | 36.60% |
| 148 | 849 | 23.61% | 42.45% | 33.95% | 55.61% |
| 149 | 827 | 30.38% | 40.56% | 29.06% | 74.89% |
| 150 | 1164 | 22.43% | 42.59% | 34.98% | 52.68% |

**Table 8. High grade meningioma features.**

| Sample | Nuclei number | Nuclei area ratio | Cytoplasm area ratio | ECS area ratio | Nuclei to cytoplasm area ratio |
|---|---|---|---|---|---|
| 1 | 1517 | 34.26% | 35.51% | 30.23% | 96.49% |
| 2 | 1235 | 18.15% | 42.35% | 39.49% | 42.87% |
| 3 | 1145 | 29.11% | 52.01% | 19.13% | 55.98% |
| 4 | 1018 | 25.92% | 35.61% | 38.47% | 72.81% |
| 5 | 1028 | 24.07% | 45.18% | 30.75% | 53.29% |
| ⋮ | ⋮ | ⋮ | ⋮ | ⋮ | ⋮ |
| 146 | 1660 | 26.44% | 36.34% | 37.22% | 72.78% |
| 147 | 1572 | 23.96% | 40.95% | 35.09% | 58.52% |
| 148 | 1758 | 34.42% | 29.91% | 35.68% | 115.09% |
| 149 | 1543 | 20.91% | 45.64% | 33.44% | 45.81% |
| 150 | 1615 | 22.28% | 43.80% | 33.92% | 50.85% |

**Table 9. Wilcox rank sum test results for the difference of high and low grade features.**

| | Nuclei number | Nuclei area ratio | Cytoplasm area ratio | ECS area ratio | Nuclei to cytoplasm area ratio |
|---|---|---|---|---|---|
| Results | 0.007948 | 0.003092 | 0.004783 | 0.3204 | 0.0001822 |

It can be seen from Table 9 that except for the proportion of ECS area ratio, the results of other features are all less than 0.05 and rejecting the null hypothesis. The reason why the proportion of ECS failed the test is that it is not the cell itself, but the effect on abnormal tissue growth is mainly reflected in the cell, and the effect on the ECS is relatively small. Therefore, the changes in pathological images of meningiomas of different grades are relatively insignificant. Therefore, on the whole, the pathological features based on the segmentation results can effectively reflect the differences in the grade of meningiomas.

## Comparison of segmentation results of public datasets

To further test the generalization performance of the proposed method, we perform segmentation experiments on the publicly available dataset MoNuSeg. The MoNuSeg dataset [27] comes from the Medical Image Computing and Computer Assisted Intervention (MICCAI) 2018 Multi-Organ Pathology Image Nucleus Segmentation Challenge, which includes 30 training sets and 14 test sets. These images come from many different hospitals and cover tumor tissue samples from many different organs, and it can be found from the following link https://monuseg.grand-challenge.org/Data/.

To enrich the number of training samples and reduce the bias of clustering results due to uneven coloring, we set a $500 \times 500$ size window to randomly grab image blocks from the original image of $1000 \times 1000$ size, and use the designed sample selection strategy on the image blocks. Table 10 shows the comparison results of different methods. From the values in the

**Table 10. MoNuSeg dataset comparison results.**

| | Method | JS | DC |
|---|---|---|---|
| Supervised | FCN [28] | 0.4935 | 0.7460 |
| | U-net [29] | 0.5942 | 0.7397 |
| | U-net++ [29] | 0.6089 | 0.7528 |
| | SegNet [30] | – | 0.7526 |
| Ours | | **0.6098** | **0.7559** |

table, it can be seen that the proposed unsupervised pseudo-label training method reaches or exceeds the accuracy of the classical supervised semantic segmentation model in terms of the accuracy of nucleus foreground segmentation. This also further proves the strong generalization ability of the proposed method.

## Conclusions

In this paper, we propose a fully automated pipeline based on pseudo-label to locate precise nuclear boundaries in HE-stained pathological images of meningiomas, which focus on the automatic generation process of pseudo-labels, and design a series of effective deep learning solutions for pseudo-labels. As an alternative to manually choosing training samples from HE image patches, an unsupervised selection strategy is proposed that can automatically and adaptively capture training samples according to their features, which greatly improves the efficiency of the training process and is compatible with various image quantities and qualities. Then, a deep learning framework is improved with strategies of full mixup and dynamic epochs in the training process. Through this framework, even incompletely stained nuclear areas can be predicted based on stably stained nuclear areas.

The proposed method shows good performance in comparison experiments against supervised semantic segmentation methods and traditional machine learning methods. For supervised semantic segmentation methods, in order to increase the generalization ability of the model, it is necessary to obtain as much as possible the labels of images with different nucleus shapes and colors, when a new case image needs to be processed, it must be manually labeled first, and then a training set is constructed to retrain the model, so that the model has the ability to segment the new case, the inevitable manual participation will greatly reduce the efficiency of image segmentation. However, the proposed method can capture training sample images and their pseudo-labels according to the established strategy. When new image case is obtained, training samples and labels can be automatically constructed based on the images for further training of the model, which improves the expansion efficiency of the model. For traditional machine learning methods, feature construction with high representation ability is a difficult problem, and the ability of pixel or superpixel classification to obtain local information of the area where the pixel is located is insufficient. The proposed method uses image patch and semantic segmentation network to solve these problems.

In addition to the above advantages, some minor problems of the proposed methods could be further improved in the future. First, segmentation of some large nuclei aggregations with regular shapes is relatively difficult. These aggregations can be mistakenly classified as a whole nucleus, and training samples with regular aggregations may be lacking. Since such aggregations are often much larger than single nuclei, the feature of nuclei size will be considered in a future training framework to distinguish such cases. Second, some tiny irrelevant dots are segmented as nuclei, which could be erased in the postprocessing after segmentation. Third, there are still some under- and over-segmented cases after using the mixed-up watershed method. Considering these possible improvements, we will further direct the unsupervised selection strategy to choose more reliable nuclear boundaries as training samples, which will improve the segmentation accuracy for the reference of more efficient clinical researches on pathological image analysis.

## Supporting information

**S1 Fig. Original pathological images of Figs 2 to 4 in the main body.**
(ZIP)

**S2 Fig. Original pathological images of Fig 5 in the main body.**
(ZIP)

**S3 Fig. Original pathological images of Fig 6 in the main body.**
(ZIP)

**S4 Fig. Original pathological images of Fig 8 in the main body.**
(ZIP)

**S5 Fig. Part 1 of original pathological images of Fig 7 in the main body.**
(ZIP)

**S6 Fig. Part 2 of original pathological images of Fig 7 in the main body.**
(ZIP)

**S1 File.**
(DOCX)

## Author Contributions

**Conceptualization:** Jing Zhong, Yunjing Xue, Peng Shi.

**Data curation:** Lin Lin, Yunjing Xue.

**Formal analysis:** Chongshu Wu, Jing Zhong, Lin Lin, Yanping Chen.

**Funding acquisition:** Jing Zhong, Peng Shi.

**Investigation:** Jing Zhong, Lin Lin, Yanping Chen, Yunjing Xue.

**Methodology:** Chongshu Wu, Peng Shi.

**Project administration:** Peng Shi.

**Supervision:** Peng Shi.

**Writing – original draft:** Chongshu Wu, Jing Zhong.

**Writing – review & editing:** Chongshu Wu, Jing Zhong, Peng Shi.

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
