## [Decision Letter · Decision Letter 0]

11 Nov 2021

PONE-D-21-22427Segmentation of HE-stained meningioma pathological images based on pseudo-labelsPLOS ONE

Dear Dr. Shi,

Thank you for submitting your manuscript to PLOS ONE. After careful consideration, we feel that it has merit but does not fully meet PLOS ONE’s publication criteria as it currently stands. Therefore, we invite you to submit a revised version of the manuscript that addresses the points raised during the review process.

We look forward to receiving your revised manuscript.

Kind regards,

Zhifan Gao

Academic Editor

PLOS ONE

Journal Requirements:

This work was supported by the Fujian Science and Technology Innovation Joint Fund 466

(2018Y9112) and the Fujian Health Science Research Talent Training Project 467

(2019-ZQN-17). 

Reviewers' comments:

Reviewer's Responses to Questions

**Comments to the Author**

1. Is the manuscript technically sound, and do the data support the conclusions?

Reviewer #1: Yes

Reviewer #2: Yes

2. Has the statistical analysis been performed appropriately and rigorously? 

Reviewer #1: Yes

Reviewer #2: Yes

3. Have the authors made all data underlying the findings in their manuscript fully available?

Reviewer #1: Yes

Reviewer #2: Yes

4. Is the manuscript presented in an intelligible fashion and written in standard English?

Reviewer #1: Yes

Reviewer #2: Yes

5. Review Comments to the Author

Reviewer #1: The proposed method and completed experiments in this paper are in line with my expectations for a manuscript on medical image processing. It would be better if there were an accepted dataset to support the experiment, can the authors provide such a supplement for this paper? In addition, some grammatical spelling may need to be a little more careful.

Reviewer #2: 1.No statistical description of the validity of the method was seen, suggesting significance.2. Comparison of images can be compared with other methods to prove the advantages of this method.The image is very good.

6. PLOS authors have the option to publish the peer review history of their article (what does this mean?). If published, this will include your full peer review and any attached files.

Reviewer #1: No

Reviewer #2: No

---

## [Author Response · Author response to Decision Letter 0]

22 Dec 2021

Reviewer #1: The proposed method and completed experiments in this paper are in line with my expectations for a manuscript on medical image processing. It would be better if there were an accepted dataset to support the experiment, can the authors provide such a supplement for this paper? In addition, some grammatical spelling may need to be a little more careful.

Answer: Thank you for your helpful comments on the manuscript. In response to this problem, we tested the performance of the proposed method on the MoNuSeg dataset, The MoNuSeg dataset comes from the Medical Image Computing and Computer Assisted Intervention (MICCAI) 2018 Multi-Organ Pathology Image Nucleus Segmentation Challenge, which includes 30 training sets and 14 test sets. The detailed results are shown in Section Ⅲ.

Reviewer #2: 1. No statistical description of the validity of the method was seen, suggesting significance.2. Comparison of images can be compared with other methods to prove the advantages of this method. The image is very good.

Answer: Thank you for your suggestions, they are very helpful. For this, we selected 150 high-grade and low-grade meningiomas with HE staining pathology images, a total of 300 images, and performed cell nucleus segmentation according to the proposed method and counted relevant pathological features. Then, the two groups of features of pathological images of high- and low-grade meningiomas were tested for differences to verify the auxiliary effect of pathological features based on the segmentation results on meningioma grading. Details can be found in Section Ⅲ. Since the meningioma dataset used only has nucleus foreground annotations and no internucleus segmentation annotations, the comparison with other methods can only give a comparison chart of the nucleus foreground segmentation results, as shown in section Ⅲ.

---

## [Editor Report · Decision Letter 1]

11 Jan 2022

Segmentation of HE-stained meningioma pathological images based on pseudo-labels

PONE-D-21-22427R1

Dear Dr. Shi,

We’re pleased to inform you that your manuscript has been judged scientifically suitable for publication and will be formally accepted for publication once it meets all outstanding technical requirements.

Kind regards,

Zhifan Gao

Academic Editor

PLOS ONE
---

## [Editor Report · Acceptance letter]

19 Jan 2022

PONE-D-21-22427R1 

Segmentation of HE-stained meningioma pathological images based on pseudo-labels 

Dear Dr. Shi:

I'm pleased to inform you that your manuscript has been deemed suitable for publication in PLOS ONE. Congratulations! Your manuscript is now with our production department. 

Kind regards, 

on behalf of

Dr. Zhifan Gao 

Academic Editor

PLOS ONE